

# A mutualism without honeydew: what benefits for *Melissotarsus emeryi* ants and armored scale insects (Diaspididae)?

Christian Peeters[1], Imre Foldi[2], Danièle Matile-Ferrero[2] and Brian L. Fisher[3]

[1] Institut d'Écologie et Sciences de l'Environnement, Sorbonne Universités UPMC, Paris, France
[2] Institut de Systématique, Évolution, Biodiversité, Muséum national d'Histoire naturelle, Paris, France
[3] Department of Entomology, California Academy of Sciences, San Francisco, CA, United States of America

## ABSTRACT

Mutualisms between ants and sap-sucking insects generally involve clear benefits for both partners: the ants provide protection in exchange for honeydew. However, a single ant genus associates with armoured scale insects (Diaspididae) that do not excrete honeydew. We studied three colonies of *Melissotarsus emeryi* ants from two localities in Mozambique. Vast numbers of the diaspidid *Morganella conspicua* occupied galleries dug by the ants under the bark of living trees. Unlike free-living *M. conspicua* and other diaspidids, *M. conspicua* living with ants are known to lack shields, likely because they gain protection against enemies and desiccation. Nevertheless, we documented the occurrence of rare individuals with shields inside ant galleries, indicating that their glands continue to secrete wax and proteins as building material. This is likely to constitute a significant portion of the ants' diet, in addition to diaspidid exuviae and excretions from the Malpighian tubules. Indeed, *Melissotarsus* workers cannot walk outside the galleries due to modified middle legs, forcing them to obtain all nourishment within the tree. *Melissotarsus* founding queens, however, must locate a suitable host tree while flying, and acquire diaspidid crawlers. This mutualism involves ants that are highly specialised to chew through living wood, and diaspidids that can also live freely outside the bark. It is extremely widespread in Africa and Madagascar, recorded from 20 tree families, and harmful effects on plant hosts require rapid study.

## INTRODUCTION

Mutualisms with sap-sucking insects are one of the many fascinating adaptations that have elevated ants to pre-eminent ecological importance. Ants cannot eat fresh leaves (leaf-cutting species in the New World use a specialized fungus to process living plant tissue), but they can feed on plants via parasites that extract plant fluids directly, in a manner analogous to blood-sucking animals. Thousands of ant species qualify as herbivores due to their partnership with sap-feeding aphids and scale insects (*Gullan, 1997*; *Delabie, 2001*; *Blüthgen, Mezger & Linsenmair, 2006*). Scale insects are inherently vulnerable to predators and parasitoids because they cannot quickly withdraw their elongated sucking stylets to escape. For this reason, they can benefit greatly from ant protection, with much variation

Corresponding author
Christian Peeters,
christian.peeters@upmc.fr

in the degree of interdependence (*Gullan, 1997*; *Grimaldi & Engel, 2005*). Numerous ant species merely guard and gather honeydew from insects that live outside of ant nests (*Buckley & Gullan, 1991*). Other species locate scale insects and build carton shelters to protect them against predators, parasites, and the weather (*Delabie, 2001*). More specialized symbioses involve tropical ants and Pseudococcidae (mealybugs) or Coccidae (soft scales) sharing the same nests. In Southeast Asia, 12 species of *Dolichoderus* ants herd 36 species of mealybugs, which the ants transport to new plant growth and shelter in their own nests (*Maschwitz & Hänel, 1985*; *Dill, Williams & Maschwitz, 2002*). When the mealybugs require better feeding sites, the entire colony relocates. In contrast to such nomadic habits, many other ants keep scale insects permanently in their domatia (i.e., specialized plant structures that house ant colonies) (*Grimaldi & Engel, 2005*). For example, *Azteca* ants living in *Cecropia* trees line all hollow branches with mealybugs (*Longino, 1991*). In SE Asia, *Macaranga* plants, *Crematogaster* ants, and soft scales depend on one another for survival (*Heckroth, Fiala & Maschwitz, 1999*; *Ueda et al., 2008*). *Acropyga* are subterranean ants that dig their nest galleries around mealybugs feeding on superficial roots (*Delabie, 2001*; *LaPolla, Cover & Mueller, 2002*). Similarly, three species of *Pseudolasius* ants inhabiting leaf litter keep different developmental stages of eight species of mealybugs in nest chambers, away from roots (*Malsch et al., 2001*). Ants carry the mealybugs to suitable roots, and bring them along during nest relocations. Among families of scale insects, the Diaspididae (armored scale insects) are considered more highly evolved (*Miller, 1990*) and include more species (about 2,700) than the Pseudococcidae (2,500) or the Coccidae (1,500 species), yet just one ant genus practices mutualisms with diaspidids. The singular association with *Melissotarsus* ants is a puzzle since diaspidids excrete no honeydew.

Unlike soft scales and mealybugs, diaspidids do not feed on phloem sap, but rather inject saliva containing a potent mix of enzymes into meristem tissue (*Foldi, 1990a*). Predigested food is then sucked up. Moreover, their gut is discontinuous and lacks the filter chamber typical of honeydew-excreting scale insects (in the latter, excess sugar and water in the anterior midgut are directed by osmoregulation to the posterior midgut and rectum) (*Foldi, 1990a*). All diaspidids are legless (except for first instars) and build a hard protective shield using their own secretions (*Foldi, 1990b*), hence the name armored scale insects. Other families of scale insects are also capable of building wax protections, but these are less elaborate than in diaspidids (*Foldi, 1997*). Importantly, diaspidids living with *Melissotarsus* ants have never been found with wax shields (*Delage-Darchen, Matile-Ferrero & Balachowsky, 1972*; *Ben-Dov & Matile-Ferrero, 1984*; *Mony et al., 2007*). Although widely distributed throughout Africa and Madagascar, *Melissotarsus* ants (subfamily Myrmicinae) are little known due to their extremely cryptic habits. They live beneath the bark of many different species of trees, and even infest orchards (*Delage-Darchen, 1972*; *Mony et al., 2002*; *Mony et al., 2007*; *Ben-Dov & Fisher, 2010*). Four ant species (*Bolton, 1982*) are obligately involved with at least 13 species of Diaspididae, but some of the latter are also known to live freely, protected by the shields they build (*Delage-Darchen, Matile-Ferrero & Balachowsky, 1972*; *Ben-Dov & Fisher, 2010*; *Ben-Dov, 2010*; *Schneider et al., 2013*).

In the mutualism between *Melissotarsus emeryi* and *Morganella conspicua* in the Western Cape of South Africa, it was claimed that the diaspidids are reared for their meat
(*Ben-Dov, 2010*; *Schneider et al., 2013*). We studied the association between the same partners in Mozambique, over 2,000 km away. We lack direct observations of feeding behavior, but our discovery of rare shields inside ant nests indicates that the secretory glands of the diaspidid partners remain functional. This is indirect evidence that ants feed on the proteins and wax otherwise used to build shields, and this may be the main trophic benefit. We review various aspects of the biology of free-living diaspidids, including shield building (*Foldi, 1982*; *Foldi, 1990b*). *Morganella conspicua* with shields are found on the bark of ant-inhabited trees (*Prins, Ben-Dov & Rust, 1975*; *Schneider et al., 2013*), and this gives an insight into the adaptations and benefits underlying the mutualism with ants. We also discuss aspects of the life history of *Melissotarsus*, including host selection and colony founding by solitary mated queens.

## MATERIALS AND METHODS

### Taxon sampling and behavioural observations

Three colonies of *Melissotarsus emeryi* were sampled in northern and central Mozambique: (1) Namoto Forest, south of the Rovuma River and north of Quionga and Palma, Cabo Delgado province ($-10.59122°$, $40.45708°$; March 2016; two colonies); and (2) Gorongosa National Park, Sofala province ($-18.99142°$, $34.35275°$; August 2016). Inhabited trees were identified by veinlike markings on the bark, revealing the presence of galleries under the surface (*Prins, Ben-Dov & Rust, 1975*; *Schneider et al., 2013*). In Namoto, ants inhabited *Olax dissitiflora* (Olacaceae) trees of various sizes, as well as the mangrove *Xylocarpus granatum* (Meliaceae). Five sections of *O. dissitiflora* stems (50–60 cm long, 7 cm diameter) were sawed from one large tree after checking for the presence of ants, and taken to Paris (iEES) for observation. The second colony inhabited a shorter branch of a younger *O. dissitiflora* tree. See *Timberlake et al. (2013)* for discussion of the dry forest habitat south of the Rovuma River. In Gorongosa, one section of stem from *Piliostigma thonningii* (Fabaceae) was collected.

Keeping the extremities of stem sections moist allowed us to keep both diaspidids and ants alive for more than two months. Previous attempts to rear ant colonies away from their wood substrate have been unsuccessful (*Mony et al., 2007*), largely because the sap-sucking diaspidids must attach to and feed on suitable wood. Because ants will immediately spin silk to close off breaches, using sawdust gathered inside the galleries, direct observations of natural interactions with diaspidids are not possible, as mentioned previously (*Delage-Darchen, 1972*; *Mony et al., 2007*; *Schneider et al., 2013*). Over a period of several weeks, observation bouts of 1–3 h began by scraping away the outer layers of wood with a knife to expose galleries. All life stages were documented with photographs.

### Morphological study of ants and diaspidids

Specimens of diaspidids were slide-mounted and identified with a compound microscope. The alimentary canal of ant workers and larvae was dissected and slide-mounted.

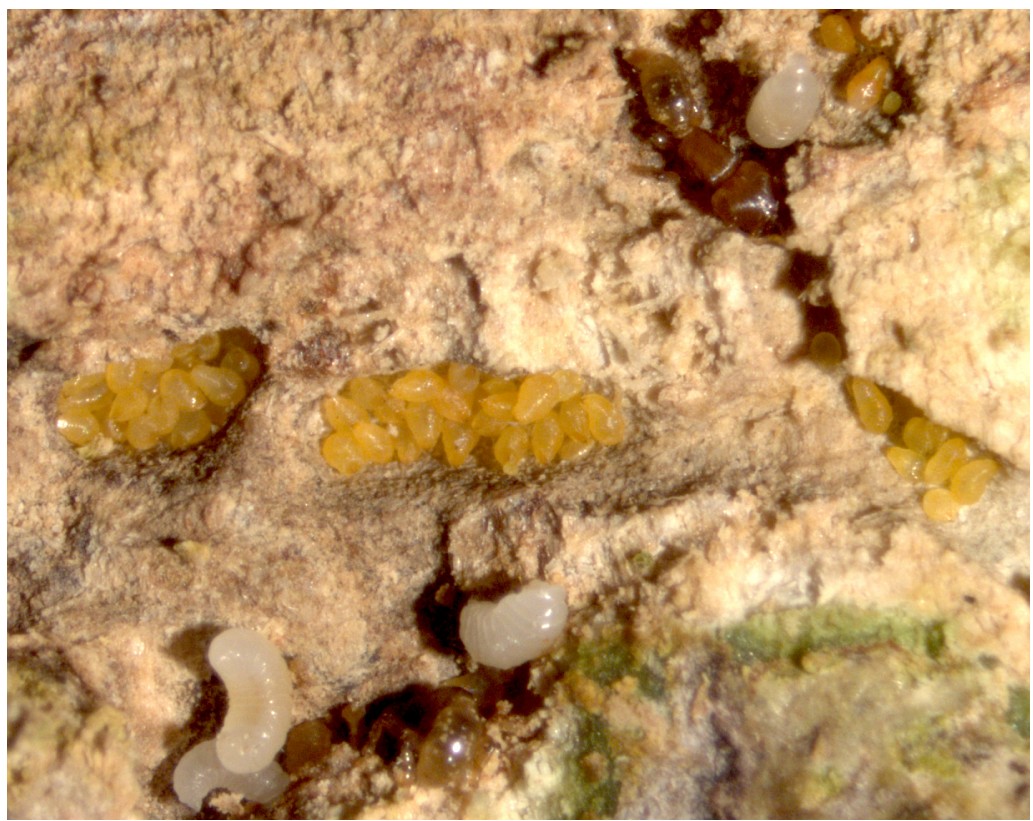

**Figure 1** Dense aggregations of adult *Morganella conspicua*, ant workers and scattered ant larvae.

## RESULTS

### *Morganella conspicua* and *Melissotarsus emeryi* in Mozambique

Thousands of diaspidids were distributed throughout the extensive network of galleries dug under bark by *M. emeryi* ants. Dense aggregations of fully-grown diaspidids occurred whenever galleries widened (Fig. 1). In *Olax dissitiflora*, some diaspidid aggregations were aligned along veins of greenish plant tissue, suggesting that the ants do not dig at random. We identified the diaspidids as *Morganella conspicua* (Brain), making it the same association that was studied in South Africa (*Prins, Ben-Dov & Rust, 1975*; *Schneider et al., 2013*). *Morganella conspicua* can also live freely on the bark of 10 tree families (*García Morales et al., 2016*), including ant-inhabited trees. We never found male diaspidids inside ant nests and reproduction is likely to be parthenogenetic, as with many diaspidids (*Koteja, 1990*).

Aggregations of *M. conspicua* included up to a few dozen fully grown females; up to 56 were counted in one chamber. While most females were yellowish, the few darker ones are the oldest, probably senescent, females. The adult females were usually crowded in ant galleries, often all parallel to each other. They "stand" with all of their pygidia facing out (Fig. 1); this is unlike free-living diaspidids which generally lie under their shields against the bark. First-instar nymphs ("crawlers" (*Gullan & Kosztarab, 1997*)) have legs and were observed to walk freely. Crawlers generally need to locate suitable plant tissue to feed on

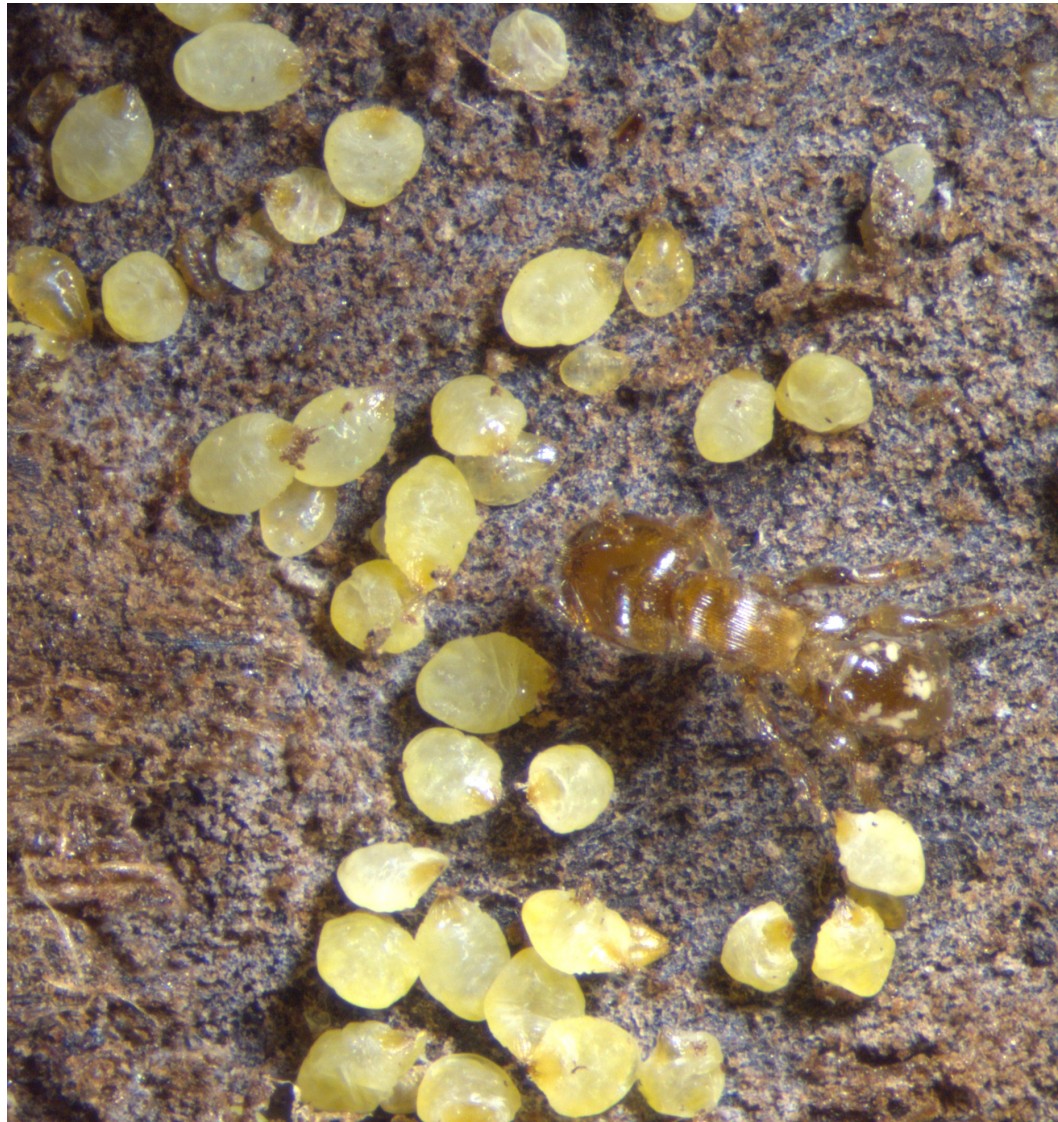

**Figure 2** Aggregation of female *Morganella conspicua* and a worker (2.5 mm long) of *Melissotarsus emeryi.*

within 24 hours, or they will die of exhaustion. Once settled, crawlers insert their stylets and start to feed, then molt into second instars and lose their legs; after this, they can never move again. Several aggregations included a range of body sizes (Fig. 2). We removed 32 fully-grown adult females from one aggregation (after several weeks in the laboratory), revealing 59 smaller diaspidids (crawlers and second instars) underneath.

We never observed ants tending or otherwise interacting with diaspidids. Upon disturbance, workers did not carry their own brood to safety. Lone ant larvae and single eggs were dispersed throughout the galleries (Figs. 1 and 3). Single pupae were scattered throughout. All sizes of larvae were found as singletons throughout the galleries, but were not seen feeding. Given that ant larvae are non-mobile, they should have been spotted

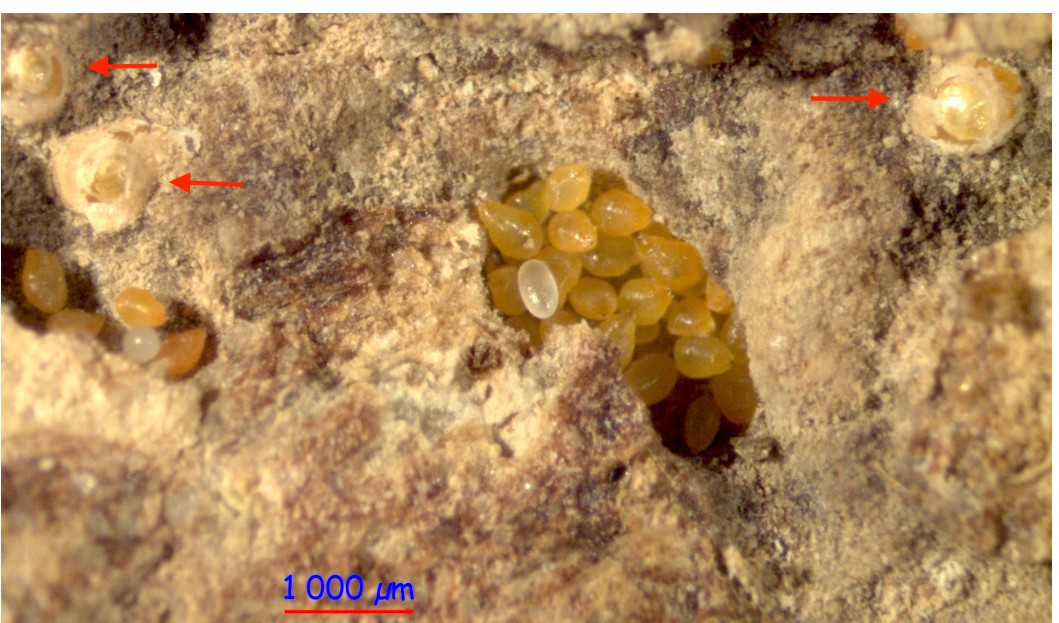

**Figure 3** Three shields (red arrows) of *Morganella conspicua* next to aggregations of naked females. Note single ant eggs scattered among the adult diaspidids.

next to partly-eaten diaspidids if diaspidid "meat" was their regular food, but this was not the case. Many ant sexuals (queens and males) were collected in the Namoto colonies. A few dealated queens were dissected: their spermatheca was full of sperm, but their ovaries were mostly inactive. However, in the Gorongosa colony, one queen was found with a hugely swollen abdomen caused by stretching of intersegmental membranes. Although *Melissotarsus* queens and workers are similar in body size, such physogastry makes a functional queen highly distinctive and reflects high fecundity. A few workers ($n > 10$) had two ovarioles with up to two yolky oocytes.

### Discovery of shields built by *M. conspicua* inside ant nests

In all three *Melissotarsus emeryi* colonies, we found a very small number of *Morganella conspicua* with shields, and they were always located apart from dense aggregations (Figs. 3 and 4). Their shields were thinner than described in free-living *Morganella conspicua* (2 mm wide, with three parts: central circular zone built by first instar, middle zone by second instar, and external zone by the adult (*Balachowsky, 1948*)). A maximum of three females with a shield were found in one spot. Nymphal exuviae had been incorporated into the shields (Fig. 5). Some shields contained one to two crawlers under their mother (Fig. 5B).

Up to eight embryos were found inside each of the ten naked individuals examined. This indicates that even though fully-grown females in aggregations lack shields, they all give birth. Shields or exuviae were never seen in dense aggregations of diaspidids. In nests opened repeatedly for observations in the lab, outlying chambers vacated by ants were soon infested by fungi and mites.

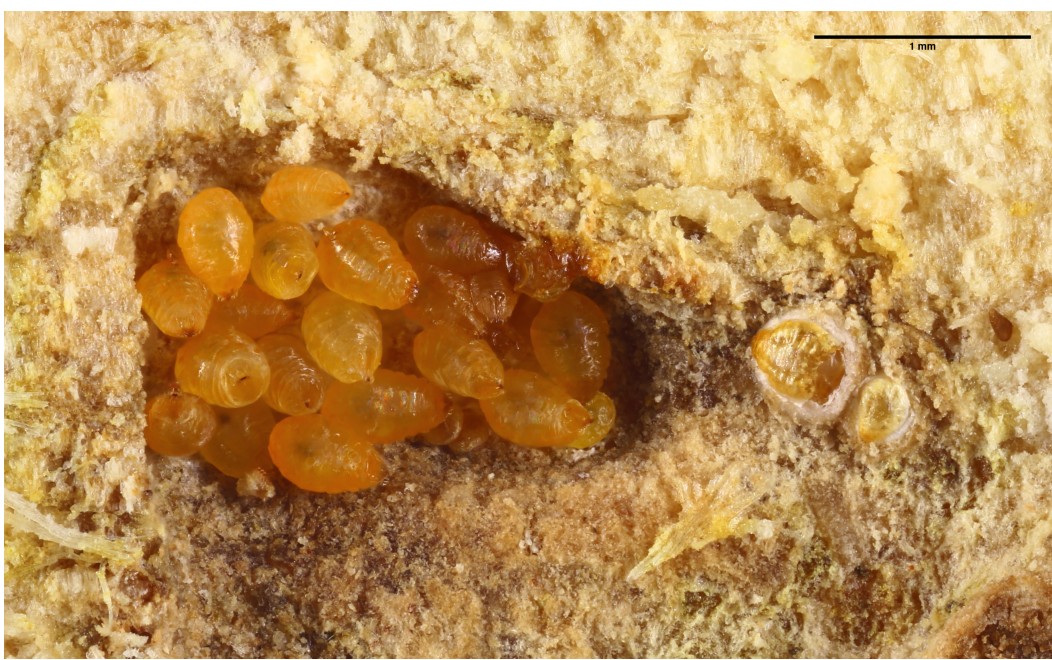

**Figure 4** Shield-less adults of *Morganella conspicua,* and two shields (on right) away from the aggregation.

## Contents of ant guts

A dark region was visible through the lightly pigmented abdomens of most ant adults, and also in larvae. The alimentary canal was dissected in at least ten workers, and examination of the midgut ("individual stomach") contents showed large quantities of a yellowish, oily substance (Fig. S1). In contrast, the crop ("social stomach") was empty, suggesting the lack of trophallaxis among nestmates. Dissection of the blind-ended guts of ant larvae showed similar yellow globules. Likewise, whole-mounts of *Morganella conspicua* revealed yellow globules throughout the body (Fig. S1). We did not identify these substances, which may already have been modified by digestive enzymes. Because direct observations of feeding behavior are impossible, we do not know what proportion of the ant diet is made up by diaspidid secretions.

## DISCUSSION

### A mutualism between "pragmatic" diaspidids and specialist ants

While *Morganella conspicua* occurs in very high densities inside ant nests, it is also known to live freely on the exterior of trees, covered with a shield like all other diaspidids (*Prins, Ben-Dov & Rust, 1975*; *Schneider et al., 2013*). In sharp contrast, *Melissotarsus* is a strictly obligate partner that is fully incapable of living or foraging outside its galleries in the host trees. Remarkably, workers cannot walk on flat surfaces due to the middle legs that project upwards and make the alternating triangle gait of insects impossible. Importantly, this genus is able to chew through living wood, and, uniquely among ants, adult workers can secrete and spin silk to seal off gaps in their galleries (*Fisher & Robertson, 1999*;

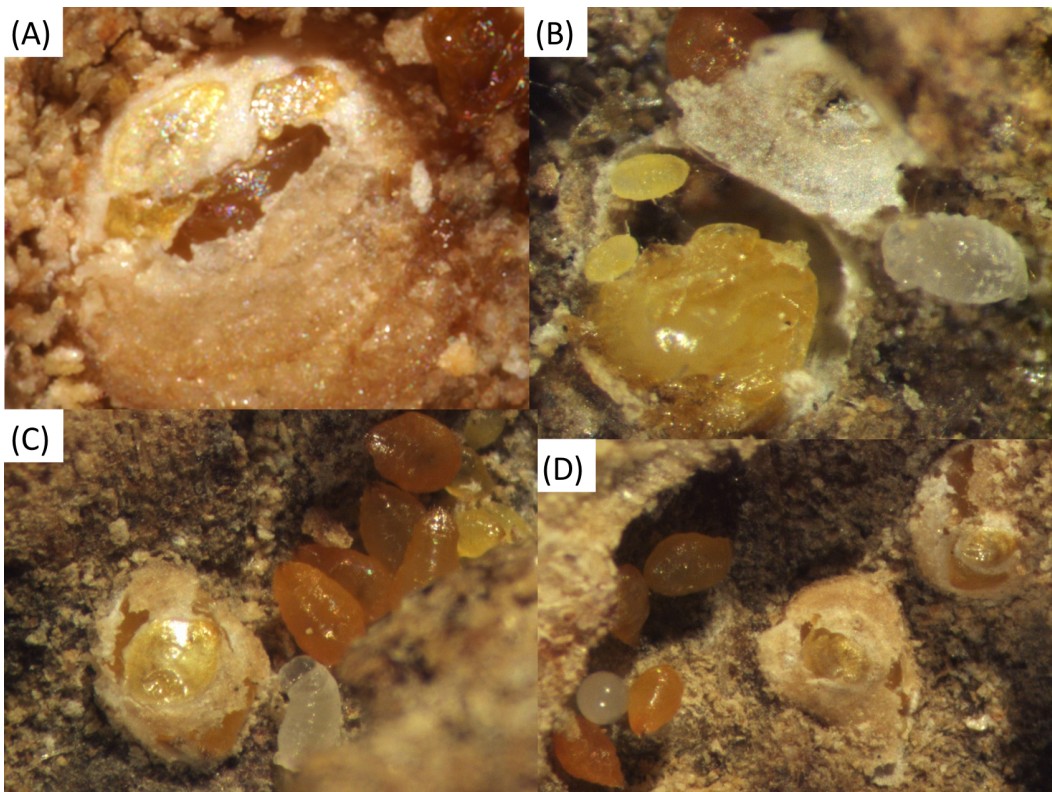

**Figure 5** **Various shields of *Morganella conspicua* inside ant galleries.** (A) Nymphal exuviae are incorporated in center of shield; (B) Shield lifted up, revealing two crawlers inside (one ant larva on the side); (C) Shield with exuvia, several shield-less adults and an ant larva; (D) Two shields next to adults and one ant egg.

*Hölldobler et al., 2014*). Indeed, as trees grow in height and diameter, bark will break open in places, and the ants must repair their galleries to keep pace.

If this mutualism is highly advantageous for *Melissotarsus*—plentiful and reliable food obtained from diaspidids, and no competition from other ants—what are the benefits for diaspidids? *Morganella conspicua* gain easy access to growing plant tissue underneath the bark, which may otherwise be largely inaccessible. They also gain protection against predators (mostly Coleoptera), parasites (e.g., chalcidoid wasps) and environmental hazards. Free-living diaspidids generally suffer huge mortality (*Foldi, 1997*) that is mitigated by building a shield. The shield is energetically expensive but protects adults and their offspring against enemies and desiccation (*Foldi, 1991*). Ant-associated diaspidids are likely to show much lower mortality than free-living species, given that the ant galleries are sealed off and defended. Thus shields seem superfluous in the benign environment of ant nests.

Free-living diaspidids generally produce 50–100 offspring, with extremes ranging from 10–600 (*Koteja, 1990*). We found embryos inside fully grown, shield-less *Morganella conspicua* females, indicating that they all reproduce and may have about 10 offspring each. The high density of adult diaspidids inside ant nests suggests an uncommonly high rate of survival, crawlers included.

## Ant-associated *Morganella conspicua* retain the ability to make shields

The lack of shields—the trademark of Diaspididae—inside ant nests constitutes a striking difference from free-living forms. However, ant-associated *M. conspicua* retain the roughly 20 secretory glands of free-living forms (*Balachowsky, 1948*), as evidenced by the presence of dorsal gland openings ("macroducts"). The occurrence of rare shields inside ant nests, newly documented here, is further evidence that the secretory glands remain physiologically active. Detailed descriptions of shield-building behavior in free-living diaspidids (*Foldi, 1982*; *Foldi, 1990b*) indicate that, after the stylets are inserted and feeding begins, wax glands secrete double-stranded filaments (0.5 μm in diameter) (*Pesson & Foldi, 1978*). Histological and cytological data have revealed that secretions include abundant glycoproteins and lipoproteins together with wax (*Foldi, 1991*). Chemical analysis confirmed that secretions are a complex mixture of free amino acids, proteins, lipids, triglycerols, terpenes, natural resins, and true waxes (*Foldi, 1991*). Filaments are deposited by the pygidium (last abdominal segments fused together) to form a mesh on the dorsum, and are progressively cemented together with anal substances (mainly glycoproteins) secreted by Malpighian tubules (*Foldi, 1982*). Nymphs are firmly attached to the substrate, and they achieve the conical or rectangular shapes of a shield by moving around the insertion point of their stylet. Soon after completion of a shield, the first instar nymph molts and its exuvia is incorporated into the shield. The second instar nymph, now legless, grows in size and continues to enlarge the shield. Its exuvia is also added to the shield (Fig. 5).

None of the shields we found were surrounded by naked females, and we suspect that living in crowded aggregations (Figs. 1 and 4) affects shield-building behavior. An experimental study (*Foldi, 1982*) showed that physical and biological factors can disturb the construction of normal shields. *Ben-Dov & Fisher (2010)* suggested that the lack of shields inside ant nests is a consequence of the ants collecting wax filaments, and we concur. Thus, ant-associated diaspidids continue to secrete wax filaments for their shields just as free-living forms do, and this is eaten by ants. Unlike *Ben-Dov (1978)*, we think the ants do not chemically inhibit the building behaviour of diaspidids, because all the shields we observed were nearby naked individuals.

## Are diaspidids "domesticated cattle"?

Although it was initially suggested (*Delage-Darchen, 1972*; *Prins, Ben-Dov & Rust, 1975*; *Ben-Dov, 1978*) that the ants benefit by collecting the waxy secretions from *Morganella conspicua*, it was later claimed (*Ben-Dov, 2010*; *Ben-Dov & Fisher, 2010*) that their meat is the primary food reward for *Melissotarsus* ants, despite the lack of direct observations of diaspidids being consumed. This claim was reasserted (*Schneider et al., 2013*) based on (i) a single observation of an ant detaching an adult diaspidid from the wood, fate unknown; and (ii) removal of all ants from one branch segment, monitoring of fully-grown diaspidids over the next 12 days, and subsequent lack of wax filaments produced (*Schneider et al., 2013*). However, our observations in three colonies show that shields are built occasionally. Moreover, after several weeks in the laboratory, as ants became fewer and had evacuated some outlying galleries, we found white deposits amid diaspidid aggregations, and even
around the edges of some smaller females (Fig. S2). Earlier reports described a coat of white matter on the walls of galleries, and this was called silk (*Schneider et al., 2013*; *Prins, Ben-Dov & Rust, 1975*; *Fisher & Robertson, 1999*) possibly because *Melissotarsus* adults, alone among ants, have silk glands in their heads (*Fisher & Robertson, 1999*; *Hölldobler et al., 2014*). However, the white deposits are visually distinct from the discrete and thicker silken strands spun by ants (Fig. S3), that are restricted to the repair of galleries. We argue that the wax filaments secreted by diaspidids are habitually consumed by the ants. Ant larvae were found scattered in the galleries (Fig. 2) and probably feed individually on secretions from *Morganella conspicua*, akin to what is known in fungus-eating ants where larvae are distributed throughout the fungus gardens (*Schultz & Brady, 2008*). Since constituents of plant tissue ingested by free-living diaspidids are used in shield formation (*Gullan & Kosztarab, 1997*), the wax secretions are akin to honeydew.

Published studies on ant-associated diaspidids have considered shields to be made solely of wax. Proteins, however, are the major constituents of shields (*Dickson, 1951*; *Foldi, 1990a*; *Foldi, 1991*; *Pesson & Foldi, 1978*), secreted from dozens of "wax" glands as well as from Malpighian tubules connected to the rectum (*Foldi, 1990a*). In addition, the Malpighian tubules in diaspidids have normal excretory functions, hence nitrogenous wastes collected from the hemolymph and released at the rectum may be another valuable source of nutrients for the ants. It is also likely that ant workers and larvae eat the exuviae of diaspidids (Fig. S4), just as many ant species eat the exuviae of their own larvae, likely to recycle nitrogen. Indeed, *Morganella* individuals were always remarkably clean and exuviae were never found.

Occasional predation has been reported in mutualisms involving various honeydew-emitting insects (*Gullan, 1997*; *Delabie, 2001*; *Dill, Williams & Maschwitz, 2002*), although *Macaranga*-associated ants do not eat their soft scales even when starved (*Heckroth, Fiala & Maschwitz, 1999*). *Melissotarsus* ants may eat *Morganella conspicua* sporadically, but none of our data substantiate the predominance of predation. Estimates in Cameroon reached 1.5 million ants (larvae included) and half a million diaspidids in a single tree (*Mony et al., 2002*). Our observations on Mozambique populations confirm that hundreds of chambers are packed with *Morganella* adults. It would be difficult to amass such a large population of fully grown diaspidids if they are habitually eaten. Instead, the ants appear to treat the diaspidids as a standing herd that allows sustainable gathering of tiny amounts of secretions from each individual. In addition, dead or dying diaspidids are likely to be eaten by the ants, because we never found corpses. The closest analogy to "domesticated cattle" known in ants are oribatid mites (*Aribates javensis*) that are obligate inhabitants in nests of *Myrmecina* sp. in Indonesia; dead mites are immediately eaten by the ants (*Ito & Takaku, 1994*).

## Diaspidids are not specialized symbiotic partners of ants

Unlike mealybugs and soft scales that can detach their stylets from the substrate and be carried around by ant partners (*Delabie, 2001*; *Maschwitz & Hänel, 1985*; *Dill, Williams & Maschwitz, 2002*; *Ueda et al., 2008*; *Gullan, Buckley & Ward, 1993*), the permanently sessile condition of diaspidids (crawlers excepted) is a serious constraint on would-be mutualisms

with ants. Thousands of scale insects (especially mealybugs and soft scales) and other honeydew-excreting insects evolved trophobiosis with ants multiple times independently (*Johnson et al., 2001*). These associations show all stages of development ranging from very loose, facultative and nonspecific, to highly coevolved, specific and obligate partnerships (*Malsch et al., 2001*; *Blüthgen, Mezger & Linsenmair, 2006*). A minority of scale insects are obligate mutualists and exhibit fascinating behavioral, physiological, or morphological adaptations. Mealybugs usually build loose waxy covers to protect their body and eggs, but the species involved in mutualisms with *Acropyga* ants delegate parental care of eggs to their host, and the mealybug young are tended together with the ant brood (*Delabie, 2001*; *LaPolla, Cover & Mueller, 2002*).

Other possible reasons why diaspidids are involved with only one ant genus are the protection gained from the shields in free-living species, and the lack of honeydew as reward for potential ant partners. Besides *Morganella conspicua,* a few diaspidid species are reported to be obligate partners (free-living forms are unknown) with *Melissotarsus* ants, but their biology has never been studied (*Ben-Dov, 2010*). We simply know that *Melanaspis madagascariensis* and *Morganella pseudospinigera* have shields when free-living, but are naked in the galleries of *Melissotarsus* species (*Delage-Darchen, Matile-Ferrero & Balachowsky, 1972*; *Ben-Dov, 2010*). In three species of the genus *Melissoaspis,* external morphology indicates reduction of secretory glands, i.e., no "macroducts" essential to build shields, as well as only few "microducts" along the ventral margins (*Ben-Dov, 2010*; *Schneider et al., 2013*). If diaspidids that cannot live freely (e.g., *Melissoaspis*) lose their secretory glands, it is not clear how they might benefit ants. Anus excretions (from the Malpighian tubules) are one option, otherwise the ants are forced to consume their partners.

Besides excavating galleries in the host tree and repairing breaches promptly, *Melissotarsus* workers may help the diaspidids with hygiene: antimicrobial secretions from their metapleural glands may be applied (*Yek & Mueller, 2011*). However, ants cannot regulate the spatial distribution of diaspidids since these never move again after inserting their stylet in plant tissue. Hence it is the crawlers that walk and spread around ant galleries. In contrast, *Melissotarsus* founding queens must play a crucial role in bringing diaspidid crawlers to new nests. Winged queens have never been collected with a diaspidid crawler in their mandibles, unlike what occurs in all species of *Acropyga* and in *Tetraponera binghami*, where young queens carry gravid mealybugs in their mandibles when they fly away from natal colonies to mate with foreign males and found new colonies (*Buschinger, Heinze & Jessen, 1987*; *Klein et al., 1992*; *Johnson et al., 2001*; *LaPolla, Cover & Mueller, 2002*). Having a symbiont from the start means that a foundress has a secure honeydew supply to feed her first generation of workers, eliminating the need to forage outside. Young queens of *Aphomomyrmex afer* disperse with a second instar female mealybug clinging to their body, ensuring the mealybug will not be a hindrance when queens chew an entrance hole into the domatium of a host plant (*Gaume, Matile-Ferrero & McKey, 2000*). Yet such vertical transmission (co-dispersal of both ant and coccoid) is relatively infrequent among mutualists, and coccoids commonly disperse independently from ant partners (i.e., horizontal transmission) (*Moog et al., 2005*). In *Cladomyrma* ants living in mutualistic plants, founding queens do not forage outside (i.e., colony foundation is claustral

(*Peeters & Molet, 2010*)), and mealybug crawlers find their own way into nest chambers inside hollow twigs (*Moog et al., 2005*). Although we cannot exclude the possibility that free-living crawlers of *Morganella conspicua* walk inside the chambers chewed by *Melissotarsus* founding queens, the latter may take a more active role. Given the ubiquity of free-living *Morganella conspicua* across Africa and Madagascar (*García Morales et al., 2016*), we suggest that flying ant queens select host plants on the basis of presence of *M. conspicua* on the bark. Soon after digging their initial shelters, foundresses bring crawlers inside or crawlers walk in on their own. Ant workers cannot be involved since they cannot walk on the bark. In a newly founded colony of *M. beccarii* under bark, a dealate queen occurred together with a nanitic worker and six diaspidids (*Delage-Darchen, 1972*). Other incipient colonies of *M. beccarii* and *M. weissi* consisted of a mated foundress, brood, fewer than 10 workers and diaspidids (*Mony et al., 2007*). Winged sexuals of *Melissotarsus beccarii* and *M. weissi* are produced year-round (*Mony et al., 2002*). After mating, a proportion of queens seem capable of dispersing over long distances, but others may return to their natal colony where some lay eggs and apparently distribute themselves throughout the host trees (*Mony et al., 2002*). Colonies have multiple highly fecund queens, and it was suggested that incipient colonies can merge with established colonies on the same tree (*Mony et al., 2007*).

## Adaptations of diaspidid-associated ants

The genus *Melissotarsus* is a morphological anomaly among ants. Among several modifications of the legs of workers are enlarged coxae and upward-pointing second pair of legs (*Fisher & Robertson, 1999*) that afford the ants better traction in the galleries by making contact with the roof. Consequently, the workers cannot forage outside the trees. It is important to remember that the queens lack this leg modification, consistent with their brief period of activity outside trees during mating and colony foundation.

The miniature workers of *Melissotarsus* are matched in size with their diaspidid partners. A majority of ant species show extreme reduction in the body size of workers, and this miniaturisation is possible because they are wingless (*Peeters & Ito, 2015*). Miniature workers are cheaper and allow considerable increases in colony size. A numerous labor force affords many benefits that are distributed in space, hence highly appropriate for tasks associated with mutualisms, such as guarding plants or gathering honeydew. Despite miniaturization, *Melissotarsus* workers are extremely strong. The heads of workers are enlarged ventrally and packed with muscles exhibiting complex geometry (J Billen & C Peeters, 2017, unpublished data). *Melissotarsus* belongs to the very small number of ants able to chew through living wood (*Fisher & Robertson, 1999*). The use of mandibles as specialized chewing tools is confirmed by the strikingly worn mandibles of older workers (*Mony et al., 2007*; *Delage-Darchen, 1972*). Workers in all species of *Melissotarsus* show obvious variations in body size (*Mony et al., 2002*; *Bolton, 1982*), and larger workers may have even more powerful mandibles to excavate harder wood. This tunneling ability contrasts with most ants nesting in live plant tissues, which often use preformed cavities or galleries dug by beetle larvae, and supply building materials such as soil (*Peeters & Wiwatwitaya, 2014*) or mulch from outside.

Is any other food available for the ants inside the galleries? *Melissotarsus* workers were shown to have the necessary enzymes to degrade plant polysaccharides, oligosaccharides, and heterosides (*Mony et al., 2013*). As workers chew and tunnel their way through the host trees, they may be capable of consuming fluids and tiny particles from the plant cytoplasm. Further studies are needed to confirm that nesting and feeding behaviors are intertwined (*Mony et al., 2013*), and to determine the carbohydrate:protein balance in the diet of *Melissotarsus.*

## Any benefits for host trees?

Many mutualisms between ants and plants involve soft scales and mealybugs as a third partner (*Heckroth, Fiala & Maschwitz, 1999*; *Gaume, Matile-Ferrero & McKey, 2000*; *Blüthgen, Mezger & Linsenmair, 2006*; *Ueda et al., 2008*). Although scale insects exploit plant sap, they provide benefits for the plant if such stable source of honeydew attracts ants that protect the plant against herbivores and/or competing plants. In this way, the host plant gains a benefit that exceeds the cost inflicted by coccoids. *Melissotarsus* and its diaspidids appear to be an exception among tripartite associations. Given that galleries excavated by ants extend throughout a tree (in Gorongosa, we checked that a tree of *Piliostigma thonningi* was occupied by ants from 50 cm to 12 m above ground), the huge number of diaspidids present must inflict a substantial cost to the plant host. Yet *Melissotarsus* ants bring none of the benefits common in other ant-plant associations. Indeed, workers have an unsteady gait when placed on the bark, hence they cannot patrol the outside of trees. Moreover, they are almost blind and their sting is very reduced and probably non-functional (*Bolton, 1982*).

Armored scale insects are among the most damaging and least understood of the pests that parasitize forest trees, fruit crops and ornamentals (*Miller & Davidson, 2005*). Diaspidids that feed on and under bark cause reduced vigor, diebacks, bark splitting etc., although symptoms are less evident than when they feed on fruits and leaves (*Kosztarab, 1990*). Ant-associated *Morganella conspicua* have considerably higher densities than those of free-living diaspidids that encrust branches and stems, hence they are likely to impact the host plants heavily by extracting resources and injecting toxins. The harmful effects of ants damaging the live bark of host trees also deserve more study. In South Africa, *Melissotarsus* was recorded as a pest on two species of *Ficus* that were infested up to the highest branches (*Prins, Ben-Dov & Rust, 1975*; *Prins, Robertson & Prins, 1990*; *Ben-Dov, 1978*). Trees in city parks were in poor condition and branches with extensive ant galleries could break off. In Cameroon, a survey conducted on 185 safoo (*Dacryodes edulis*) and 513 mango (*Mangifera indica*) trees indicated that most old, large trees are infested (*Mony et al., 2002*). *Melissotarsus* is known from southern Saudi Arabia to South Africa (*Bolton, 1982*) and Madagascar. Records of association so far include 20 tree families (Meliaceae and Olacaceae (this study) are new records) (*Ben-Dov & Fisher, 2010*). *Melissotarsus* may prove to be an extraordinarily common and widespread ant in Africa.

## CONCLUSIONS

Save for brief periods of sexual activity and colony founding by queens, *Melissotarsus* ants live in splendid isolation from the outside world. Their anomalous morphology (head

and legs) represents extreme adaptations (tunelling through live wood and silk spinning) for their obligate mutualism with diaspidids. In sharp contrast, *Morganella conspicua* continue to live both with and without ants, and their morphology is not modified. In addition to gaining access to underlying meristem tissue, they benefit from highly effective protection (enemies and weather) by the ants. Accordingly, diaspidids stop building shields even though they continue secreting wax and proteins as construction material. These secretions, together with exuviae and anal excretions, can provide an adequate array of nutrients for the ants. Furthermore, ants can also feed on dead or dying diaspidids, and possibly on plant tissues. Existing literature on all four species of *Melissotarsus* reveals similar habits of relying on diaspidids for food within the confines of galleries chewed in healthy wood. A close relationship with the Asian genus *Rhopalomastix* is evident (*Bolton, 1982*), but almost nothing is known on its biology except that most of a dying tree in Indonesia was riddled with ants burrowing in the bark, and occurrence of ant pupae suggests that it was a nest (*Wheeler, 1929*). Future studies must investigate the impact of these highly successful mutualists on host trees.

## ACKNOWLEDGEMENTS

We thank Flavia Esteves and the Malagasy Arthropod Team (Jean-Claude Rakotonirina, Jean-Jacques Rafanomezantsoa, Chrislain Ranaivo, Clavier Randrianandrasana, and Bemaheva Fidelisy) for fieldwork. We are grateful to Susanne Renner for constructive comments on an earlier version. Maridel Fredericksen kindly climbed 12 m up a tree in Gorongosa during Ant Course 2016. Laurent Fauvre and Adam Khalife took many of the stacked photographs. Romain Péronnet cared for the live ant colonies in Paris. Adam Khalife helped with some laboratory observations. Johan Billen carried out histological sections of ant heads.

### Funding

This work was supported by the French National Research Agency (ANTEVO ANR-12-JSV7-0003-01). The funders had no role in study design, data collection and analysis, decision to publish, or preparation of the manuscript.

### Grant Disclosures

The following grant information was disclosed by the authors:
French National Research Agency: ANTEVO ANR-12-JSV7-0003-01.

### Competing Interests

The authors declare there are no competing interests.

### Author Contributions

- Christian Peeters conceived and designed the experiments, performed the experiments, analyzed the data, wrote the paper, prepared figures and/or tables, reviewed drafts of the paper.

- Imre Foldi performed the experiments, reviewed drafts of the paper.
- Danièle Matile-Ferrero performed the experiments, analyzed the data, reviewed drafts of the paper.
- Brian L. Fisher wrote the paper, reviewed drafts of the paper, funding and field logistics.

### Field Study Permissions

The following information was supplied relating to field study approvals (i.e., approving body and any reference numbers):

Authorization to collect and export ants was issued by Ministério da Terra, Ambiente e Desenvolvimento Rural, Republica de Moçambique.

### Supplemental Information

Supplemental information for this article can be found online at http://dx.doi.org/10.7717/peerj.3599#supplemental-information.

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
