# Peer review of "A mutualism without honeydew: what benefits for Melissotarsus emeryi ants and armored scale insects (Diaspididae)?"

_PeerJ, doi:10.7717/peerj.3599_

## Round 0.1 · original submission · Minor Revisions

· Academic Editor

Minor Revisions

As you'll see below, the comments by the reviewers tended to be very positive, but indicated a few issues that need to be addressed before the manuscript can be accepted, as well as some interesting suggestions.

Reviewer 1 ·

Basic reporting

The manuscript is very well written and includes a complete review of the current knowledge of Melissotarsus-diaspidis association. The authors manage to combine all the current information about the system and other related ant-hemipteran associations with the perspective of their new naturalistic observations, putting together a very robust description of the natural history of Melisotarsus-Diaspididae association. Although the authors did not perform behavioral experiments or chemical analysis, they provides the study with outstanding visual evidence of their observations in the form of high quality pictures (However, see comment on fig 3 on the possibility of adding arrows to the pictures to direct to interesting parts of the image).

Experimental design

The study is a very interesting research about the natural history of a interestingant-hemipteran association. I consider the study very relevant and an important contribution. However, the research questions are only indicated in the last paragraph of introduction in a very general way (Lines 100-101). The authors should detail clearly at the end of introduction which aspects of the association were the focus of their observations and discussions. There are not experimental procedures in the research. Must evidence is in the form or valuable and novel (although circumstantial) natural history observations.

Validity of the findings

The authors put together a lot of valuable information to describe several aspects of the Melisotarsus-Diaspididae association. The study have no statistical analysis and the discussion is based on observational evidence. Although the presented evidence comes only from three ant nests, their findings are significant for the understanding of the described mutualism and very well putted in the context of the natural history of the species. However, I believe that authors must specify in the discussion the potential of generalization of their findings and the possible limitations derived from the limited sample size.
As one of the main objectives of study the authors present a strong case to defend the hypothesis that ants consume wax secretions form diaspids as food source and that this is the reason behind the lack of wax shield of the diaspids. Although I agree with most of the reasoning, I think that the evidence is circumstantial and not enough to discard completely other alternative or complementary hypotheses (see comment for lines 267-269). For this reason, I believe the authors should be more careful in some of their statements in order to recognize alternative (however less probable) explanations in the discussion (for instance sentence on lines 242-244 and 289-290). They may explain with more detail why alternative hypothesis (most of them already mentioned by the authors) should be discarded or suggest possible experimental pathways to completely disprove them.
I believe that none of the above issues invalidate the findings and main discussion of the study, and that with minor modifications those issues may be fixed.

Additional comments

Lines 74-78: I suggest to merge this lines with the nest paragraph (Lines 79-97)
Lines 98-101. You should state clearly here if Morganella conspicua is one of the diaspidis species that are known to also live freely (you put this information only in the discussion, but this is very relevant for the background of your study).
Lines 112-115: It should be a good idea to include geographic coordinates for sampling localities. Also, if possible, you should provide more details about the nest conditions on the field, such as the extend that ants and diaspids were present in each tree individual (this information is only scattered in the discussion)
Line 120: Specify where in Paris you performed the observations.
Lines 125-134: The observations of open galleries were all performed 3 months after the field collections or the stems were opened gradually?If gradually, then following which periodicity? Is it possible that ant behavior related to diaspidids would be affected by the time of lab rearing?
Line 240-242: Is it possible that the diaspids individuals bearing wax shields in your study correspond to males (as suggested by Prins et al 1975 and Schneider et al 2013)?
Line 242-244: I suggest using something such as “argue” rather than “conclude”. This may looks just as a matter of preference between words, but since you did not manage to perform chemical analysis or direct observations of the ant feeding behavior, there is always the possibility of alternative (however less probable) explanations. For instance, one can argue that ants may not eat the wax but that the constant interactions with ants inhibits the secretions of wax, and only those diaspids living in nest points with restricted access to ants develop the wax shield (this is actually derived from one of the alternative hypothesis proposed by Ben-Dov & Fisher 2010).
Line 267-269: Do you have observations or other evidence to support this claim of lack of chemical inhibition of ants? From the three alternative hypothesis by Ben-Dov & Fisher 2010, I believe that your evidence allows to discard hypothesis 1 (lack of capability of diaspids to produce shields) and give strong support to hypothesis 2 (removal and consumption of wax by ants). However, I think that it is not jet possible to discard completely hypotheses 3 (chemical interference). Also, as I mention in the last commentary, do you think that ant inhibition of wax production may be triggered also by behavioral stimuli and not just by chemical interference? Chemical or behavioral interference of ants
Line 289-290: considering that you never observed direct interaction (line 165), how can you be sure that ants “habitually” consume wax filaments?. I agree that the evidence of your study and other studies points in this direction but can you be sure without chemical analyses or behavioral experiments?
Line 675: You may add a mark or an arrow pointing to the shields in the picture. This may be also useful for other figures such as 5 and 6 to facilitate the understanding of descriptions of interesting parts of the image.

Reviewer 2 ·

Basic reporting

no comment

Experimental design

no comment

Validity of the findings

no comment

Additional comments

This is a very valuable and well-written contribution to a previously unstudied natural history of a mutualism between ants and diaspidids. Trophobiotic relationships between ants and hemipterans, mediated by sweet honeydew, are a classical text-book example of mutualism and have been widely studied, but the broad variation in these interactions has been surprisingly poorly documented so far. This study contributes a very important case, not only taxonomically (ant-attended diaspidids have been poorly documented), but even more importantly in terms of the mutual benefits (no honeydew excretions, instead it is shown that ants use the complex ‘waxy secretions’ produced by diaspidids that is otherwise used to form a protective shield). The evidence is sound, the observations are detailed and well documented, the argumentation in the discussion is well balanced, the existing literature well covered and the case is very convincing. Potential consequences and speculations are sufficiently well handled with care.

The comparison wax versus honeydew may need some further exploration. Are both produced by homologous glands – and is it thus just the chemical composition that differs? How widespread in the hemiptera or scale insects is the production of honeydew versus wax, including taxa that are closely related to diaspidids, and also within diaspidids? (See e.g. fulgoroids that produce wax, whereas related cicadas produce honeydew)

Similarly, the ant species also has a very unusual lifestyle, particularly since it seems to largely live within its self-formed wood cavities. Is this lifestyle common for this genus, tribe or larger taxonomic framework? Is it likely that there are far more similar, but undetected relationships?

One potential far-reaching consequence of this discovery, however, is surprisingly not examined further – an overlooked impact on many host trees: “The harmful effects of damaging the live bark of plant hosts deserves more study ... Melissotarsus may prove to be an extraordinarily common and widespread ant in Africa. The huge populations of diaspidids inside ant galleries are likely to have a heavy impact on crop plantations and managed forests“. A quick quantitative survey on the infestation rate of ant-diaspidid interactions in a few plantations (counting if diaspidids are involved or not, whether or not the wax shields are expressed etc.) would have helped to support this statement and further broaden the relevance of the article. Perhaps the authors have some data to test this that they can add?

---

## Round 0.2 · accepted · Accept

· Academic Editor

Accept

The authors did a good job in addressing the (minor) points raised by the reviewers. I believe this is an important contribution and I'm happy to recommend its acceptance.